# Prostate Cancer Knowledge, Beliefs and Screening Uptake among Black Survivors: A Qualitative Exploration at a Tertiary Hospital, Limpopo Province, South Africa

**DOI:** 10.3390/ijerph21091212

**Published:** 2024-09-15

**Authors:** Shai Nkoana, Tholene Sodi, Mahlapahlapana Themane

**Affiliations:** 1Department of Psychology, University of Limpopo, Polokwane 0727, South Africa; 2DSI/NRF-UL SARChl Chair in Mental Health, University of Limpopo, Polokwane 0727, South Africa; tholene.sodi@ul.ac.za; 3DSI/NRF-UL SARChl Chair in Schools as Enabling Environments, University of Limpopo, Polokwane 0727, South Africa; mahlapahlapana.themane@ul.ac.za

**Keywords:** prostate cancer, knowledge, attitudes, beliefs, screening, South Africa, black men

## Abstract

Men of African ancestry suffer disproportionately from prostate cancer (PCa) compared to other racial groups in South Africa. Equally concerning is that black South African men generally present later and with higher stages and grades of the disease than their non-black counterparts. Despite this, a small percentage of black South African men participate in screening practices for PCa. This study sought to explore knowledge and beliefs of black South African PCa survivors, and the potential impact of this on the limited screening uptake within this population group. A hermeneutic phenomenological study design was undertaken. The sample comprised 20 black South African PCa survivors, between the ages of 67 and 85 years (mean*_age_* = 76 yrs; SD = 5.3), receiving some form of treatment at a tertiary Academic Hospital, Limpopo Province, South Africa. The sample was selected through a purposive sampling method. Data for the study were collected through in-depth, semi-structured individual interviews and analyzed through interpretative phenomenological analysis (IPA). The findings demonstrated that black South African men had poor knowledge of PCa and that this may create an unfortunate system that precludes this population group from taking part in life-saving PCa screening services. The results highlight a need to elevate knowledge and awareness of PCa among black South African men and ultimately enhance screening practices.

## 1. Introduction

Prostate cancer (PCa) is a disease of increasing concern for the male population, and with a rising global incidence. The disease is rated the second most frequently diagnosed malignancy—after lung cancer—and is the sixth leading cause of death among men worldwide [1,2]. According to the Lancet Commission on PCa, the number of new disease cases annually will rise from 1.4 million in 2020 to 2.9 million by 2040, globally [3]. It has been reported that in 2020, PCa was responsible for 375,304 deaths globally, translating to 6.8% of all deaths among men worldwide [4]. The true number of cases are likely to be higher than the recorded figures because of underdiagnosis and poor reporting, especially in low-income and middle-income countries (LIMCs).

The incidence and mortality rate of PCa in the Sub-Saharan Africa region is 40.5 and 22.5 per 100,000, respectively, per year [5]. In South Africa, black men were reported to have higher overall incidence and mortality as a result of PCa than any other racial groups [4,6,7]. Previous studies [2,8,9] have demonstrated that black men in South Africa frequently have late PCa diagnosis and with a more advanced stage than other racial groups [2,5]. Some of the explanations for advanced stage at diagnosis among black South African men are attributed to socio-economic and healthcare access factors [2], illiteracy and a lack of knowledge about the disease [2,5,9] and the use of traditional and complementary medicine [10].

Earlier diagnosis is key to providing effective cancer control globally [9]. The screening of asymptomatic men for PCa in its early stages can be an effective measure to reduce the alarming rate of morbidity and mortality from the disease [8,11]. It is, however, important to make a distinction between screening and diagnostic testing. Screening refers to testing an asymptomatic (showing no or disguised symptoms) person with an increased risk of developing PCa [9], whereas diagnostic testing is intended for those showing symptoms in need of a diagnosis [12]. Diagnostic testing is beyond the scope of this study.

PCa screening may offer opportunities for an earlier diagnosis of the disease while still localized [9,11]. Screening, however, is still a controversial subject. For instance, there remains concerns of testing too frequently, at a young age, at an advanced age, or when short life expectancy precludes any survival benefits from screening [2]. PCa screening may result in the early detection of PCa, enabling more effective treatment and a better chance of recovery [2]. The most commonly used methods for screening men for PCa are prostate-specific antigen (PSA) and digital rectal examination (DRE) [13]. According to [14], PSA and DRE screening of asymptomatic men reduces PCa morbidity and mortality from the disease. The American Urological Association guidelines recommend PCa screening in men <55 years of age, based on high risk (e.g., family history or African America race) [13]. In South Africa, screening for PCa is conducted in an opportunistic manner rather than following an organized population-based systematic policy. The limitation of opportunistic screening is that not all potential men at risk are covered [2].

The World Health Organization (WHO) emphasizes the early detection and diagnosis of PCa as a priority [15]. Accordingly, the timely detection and diagnosis of PCa remain a major arsenal against advanced-stage presentation and mortality from the disease. Low screening rates (leading to underdiagnosis and undertreatment) cause unimaginable harm in LIMCs. Most research studies on PCa screening, including prevalence and benefits, have largely been conducted in majority-white populations in North America and European regions [15]. There is lack of studies on PCa screening in majority-black communities, largely in LIMCs, where there is a disproportionate burden of the disease. Despite the importance of early detection in reducing morbidity and mortality rates from PCa, results from most studies indicate a low participation of black men, particularly LIMCs, in screening programs [3,16]. Globally, low socioeconomic status [17], lack of awareness and knowledge about PCa and screening [1,16], discrepancies and inequalities in access to health [18,19] and cultural beliefs, stigma, and fear [17] have been implicated as contributing factors to black men’s poor uptake of PCa screening.

The Health Belief Model (HBM), one of the most widely used models in behavioral medicine, posits that people will take action to prevent illness if they regard themselves as susceptible to a condition (perceived susceptibility) [20]. There are limited studies that focused on knowledge of and beliefs about PCa among black South African men and its potential role in explaining why this population group is less likely to attend screening. Exploring the knowledge and cultural worldview of black men in relation to PCa will assist in the determination of their perceived susceptibility. The cultural worldview of individuals is rooted in the values, beliefs and behaviors of the ethnic groups they belong to. This emerges from a mix of individual meaning-making (personal factors) and interactions with others around them (social factors) [17]. People’s culture (including belief systems) may help to explain the way people understand and develop attitudes about illnesses, particularly debilitating, life-threatening diseases. There is currently a paucity of qualitative research that explains PCa screening behaviors using a theoretic model. Understanding black men’s knowledge of PCa and why so many do not take part in screening behaviors is, therefore, a legitimate variable to investigate.

## 2. Methods

### 2.1. Study Design

A hermeneutic phenomenological design was used. A hermeneutic phenomenological design was deemed appropriate, as the focus of the study was to explore knowledge and beliefs about PCa among black South African men. Hermeneutic phenomenology is concerned with the life, world or human experience as it is lived. Human beings are motivated to create meaning in the different experiences that shape their lives [20,21]. The focus was on how the participants in the study perceive and talk about PCa in order to understand and appreciate their knowledge and beliefs from their perspectives. The study was concerned with how things appear from the eyes of the participants.

### 2.2. Sampling

The participants for the study were identified and recruited using a purposive sampling technique. Through this technique, twenty (20) elderly black South African PCa survivors were selected for participation. All the participants were diagnosed with PCa (and were receiving some form of treatment at a tertiary hospital in Limpopo Province, South Africa) for more than five years prior to the commencement of the study. The inclusion criteria included the following considerations: black South African citizen, and able to communicate in at least one of the following regional languages: English, Sepedi, Xitsonga, or Tshivenda. The bias towards the chosen languages was because they are the ones most spoken within the province. Therefore, both theoretical and practical considerations formed the basis for the sampling choice [21,22].

### 2.3. Data Collection

In-depth, semi-structured individual interviews were conducted with each participant in the study. The data collection method was deemed appropriate to elicit a rich, detailed, case-by-case, first-person account of each participant’s narration of their PCa knowledge, beliefs and any previous PCa screening participation. An interview guide was used to elicit the participants’ individual narratives. This interview guide was first pre-tested for validity with two black PCa survivors who were in remission. All interviews were conducted in each participant’s preferred language and audio-recorded (with permission).

The researcher used a set of questions on an interview guide, with follow-up where necessary, to allow a participant-led interaction. The interview guide included the following open-ended questions: Tell me about your knowledge of the disease? Tell me what you think is the cause of the disease? Tell me if you ever previously screened for the disease prior to your diagnosis? Tell me how you cope with the disease, including the treatments? The method also gives enough space and flexibility for original and unexpected materials to emerge, into which the researcher may inquire in more detail using probing questions [23]. The interviews were conducted by the primary researcher, who is an experienced practicing clinical psychologist. The primary purpose was to capture how the participants narrate their knowledge and beliefs regarding PCa.

For broader scientific community access, all the individual interviews were transcribed in the participants’ preferred language and translated into English.

### 2.4. Data Analysis

Interpretative phenomenological analysis (IPA) was used to analyze the data collected in the study. IPA is an elaborate and inductive process that allows for unanticipated and unpredicted themes to emerge during analysis. Researchers applying IPA have two main aims: (a) to listen to the accounts expressed by each participant in order to obtain an insider’s perspective of the phenomenon under study and (b) to attempt to interpret these accounts in order to gain an understanding of what it means for each participant to have those accounts in that particular context [24]. Participants’ IPA explores how research participants make sense of their lived world [24]. For this reason, samples in IPA are relatively small to enable in-depth case-by-case analysis [21]. The IPA was conducted by the primary researcher, who is experienced in the technique.

### 2.5. Ethical Considerations

The study obtained appropriate ethical approval before commencement. The approvals were obtained from the University of Limpopo Turfloop Research Ethics Committee (TREC/26/2015), as well as the Limpopo Provincial Department of Health gate keeper permission (Ref:4/2/2). All the participants in the study signed informed consent (following a detailed description of the study) and no individual names were used (anonymity).

### 2.6. Trustworthiness of the Study

All the authors ensured that the criteria for credibility, transferability, dependability and confirmability were maintained throughout the study. All researchers cross-checked and immersed themselves in a reflective engagement with the participants’ narratives and meaning-making processes.

## 3. Results

All the participants (*n* = 20) were black South African men. Their ages ranged from 67 to 85 years (mean age = 76; SD = 5.3). All (*n* = 20) were PCa survivors diagnosed more than five years before the commencement of the study. The majority (*n* = 15) had primary education, and were in retirement (*n* = 13). All (*n* = 19) but one had no family history of cancer. All (*n* = 20) had no knowledge of PCa prior to their diagnosis, and all (*n* = 20) had no prior history of ever screening for PCa.

Two themes emerged from the IPA (see Table 1), highlighting participants’ knowledge and beliefs about PCa. To elucidate these themes, participants’ narrative extracts are presented, followed by a discussion of the overall findings.

## 4. Discussion

The aim of this study was to explore knowledge of and beliefs about PCa among black South African survivors. The exploration was based on the HBM explanatory framework or model. The HBM provided a framework for the examination of how knowledge and beliefs can impact PCa screening practices (uptake) among South African black survivors. HBM plays a significant role in predicting, explaining and modifying health behaviours, including screening practices [20]. The model posits that men will take action (e.g., screening) if they regard themselves as susceptible to developing PCa. Considering the complexity of PCa screening behaviors, individuals must first perceive the elevated risk of developing the disease and anticipate the positive outcomes they will gain by participating in regular screening uptake (perceived benefits) [20,21,22,23,24,25]. This model may offer useful insights in understanding potential facilitators and barriers for PCa screening among black South African men.

It has been noted that fundamental elements related to ethnicity and culture shape people’s health perceptions, attitudes, and behaviors [18,25,26]. This study identified poor knowledge of PCa among all the participants. Some of the participants in the study held myths and misconceptions about the causes of PCa. For example, some of the participants believed that the disease is caused by supernatural forces (e.g., witchcraft, bad luck, wrath of angry ancestors, etc.) [27,28,29]. Cultural influences on PCa knowledge and beliefs are multi-level and center on personal, social and wider structural factors [17].

All the participants in the study did not know PCa prior to their diagnosis. Meaning-making relies on language, and certain medical terms such as cancer (including PCa) are heavily implicated with fatalistic beliefs and negative outcomes. According to [17], where no local translations or everyday equivalents exist, discussions of cancer (particularly PCa) become difficult as people have no words of their own comfort to use. All the participants in the study did not have an equivalent indigenous language definition for PCa. The majority of the participants in the study had primary school education with limited English proficiency. As has been demonstrated through the participants’ narratives, descriptors referenced PCa by its effects (fatalism). The participants refenced PCa by its effects, i.e., that it is a disease that kills. Furthermore, because of the internal position of the organ, referencing the prostate may even be difficult. This led to PCa being made to be both invisible and mysterious. This is in line with what has been established in other studies [17,25,27]. Previous studies [16,18,30] have identified limited knowledge, including misconceptions, myths, and stigma surrounding PCa, as potential barriers to the low uptake of screening practices among black men. Indigenous language barriers and low health literacy are crucial elements in understanding cancer care pathways [31,32] within the study population. This may be compounded by other variables such as a lack of knowledge about the existence of available PCa screening methods, such as PSA and/or DRE [33,34]. Additionally, it may be possible that most at-risk black men, particularly from rural communities, may not even know when and where to go for screening [34,35,36].

Beliefs have also been proven to play an important role in influencing the uptake of PCa screening. There was a general belief among the participants in the study that cancer is incurable (cancer fatalism). In their majority, through their narratives, the participants seemed to perceive PCa diagnosis as a death sentence. This is in line with what has been established in other studies [17,25]. Achieving sufficient uptake and participation in PCa screening practices among black South African men will require in-depth understanding of the culture-informed prostate cancer beliefs and attitudes among this target group. Personal and societal (ethnic and cultural) factors interplay to produce meaning-making attributes that are crucial determinants of people’s health behavior, including health practices.

## 5. Conclusions

As demonstrated through the HBM explanatory framework, if black South African men have poor knowledge of PCa and/or do not believe they are at risk for developing the disease, they are less likely to engage in screening practices. Our study suggests that while the HBM is important in predicting prostate cancer screening intent, normative beliefs (represented by culture) dictate a person’s acceptance of a health behavior. Black South African men’s reasons for what they believe about PCa, including poor knowledge and lack of screening uptake, emanate from a mix of personal, social and structural factors. Our study suggests that indigenous language barriers and low literacy level play a critical role in the poor PCa screening uptake among black South African men. Understanding how these come together to create meaning for PCa is important in enhancing screening practices in this target population. Therefore, both western and indigenous definitions of PCa should be included in any programs targeting black South African men. Culturally relevant interventions are needed to address the barriers to PCa screening uptake among black South African men.

### Limitations of the Study

The authors would like to acknowledge that there are limitations to this study. Participants in the study were elderly diagnosed PCa survivors who were already receiving some form of treatment for the disease, and the results may not be generalizable to healthy (undiagnosed) South African black men. Future research should focus on the influence of age on knowledge, beliefs and screening behavior among undiagnosed black South African men. Such studies will help examine whether there is any shift in mindset across generations, which factors contribute to this and how it could potentially help to enhance PCa screening uptake in the target population. Despite this, the results in this study may further contribute to the body of literature on knowledge, cultural factors and screening intentions, which is currently lacking among black South African men.

## Figures and Tables

**Table 1 ijerph-21-01212-t001:** Quotes highlighting lack of knowledge and beliefs (fatalism) about PCa among black men.

Theme	Participants	Representative Quotes
Knowledge: The results showed that the participants had poor knowledge of PCa. In the majority, they highlighted this lack of knowledge through their quotes.	A	*“It is now three years with this illness. I did not know this illness. No…nobody told me this illness”.*
	D	*“I don’t know this. This is totally new to me, I have never seen or heard about this illness. I did not know anybody who had this illness before”.*
	F	*“No. I have never heard about this illness before. I do not know it. I don’t know how it come and I have never gone to the hospital to check before”.*
	K	*“No, there is no one in my family who was having this illness. It is new. I did not know it. I have never heard about the illness before”.*
	L	*“I only started to hear when they say I have cancer. I do not even know what is prostate. The doctor just say so…but…but… I never know this. I did not go to school”.*
Fatalism (beliefs): The results showed that the participants harbored fatalistic beliefs about cancer.	C	*“I don’t about this disease. Then I go home and they tell me this illness…it ….it is dangerous. They say it kills”.*
	Q	*“Someone say many people died….and they don’t know how to treat it”.*
	J	*“I did not know anything about the disease. The doctor said he has seen many old men like me with this disease but it was for the first time for me to hear about it. I once went to a funeral and the people there were saying the woman died of cancer that was on her breast. This is the only thing I know. They said at the funeral of the dead woman that cancer can kill you fast”.*

## Data Availability

The data used in the study are available on request from the corresponding author. The data are not publicly available due to the ethical nature of the study, which did not include data sharing.

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
