# Peer review of "Prostate Cancer Knowledge, Beliefs and Screening Uptake among Black Survivors: A Qualitative Exploration at a Tertiary Hospital, Limpopo Province, South Africa"

_ijerph, 2024, doi:10.3390/ijerph21091212_

Round 1

Reviewer 1 Report

Comments and Suggestions for Authors

The authors have conducted a qualitative study of knowledge about prostate cancer among older South African men with a history of prostate cancer.  

My general comment is that I would have liked the authors to provide a bit more detail on the methods, particularly the list of questions from the interview guide, and whether the questions changed over the course of the study. 

My specific comments:

Page 1, Line 39:  I did not find a statement of expectation of an exponential rise in prostate cancer incidence and mortality rate in reference 17.    This suggests that all the references need to be checked to make sure they are reflecting the statements in the text.

Page 1, Line 41:  The link in reference 6 does not seem to be working.

Page 2 Line 58:  The authors' statement that "Overall, screening benefits far outweigh the disadvantages" is not true for all men, depending largely on age and life expectancy, and needs qualification. 

Page 2, Line 61, the term PSA is defined as "prostate antigen serum" but is probably better termed "prostate-specific antigen". 

Comments on the Quality of English Language

There are a few typographical errors in the text and the text needs to be carefully edited.

Author Response

Attached, receive the table of corrections to the manuscript. 

Reviewer 2 Report

Comments and Suggestions for Authors

Thank you for an opportunity to review this manuscript. It is a very important study, and it is particularly valuable as it adds a new cultural and geographical perspective to the existing body of literature on prostate cancer. However, I have some comments and suggestions for the authors how they could make their study more impactful.

My main question to the authors addressed their decision to study screening uptake by looking at cancer survivors. As screening is usually recommended to persons who are at risk of developing the disease, it is not clear why screening uptake is being explored by interviewing people who are already diagnosed and treated for prostate cancer. It’s easy to assume that these individuals have very different experiences and attitudes towards the disease and screening compared to individuals eligible for screening. I suggest the authors provide some explanation about their rationale in choosing this group of patients for their study.

There is also further inconsistency in how the authors frame the goals of their study. In lines 101-102, they claim that “…the focus of the study was to unfold the meaning that the participants give to their experiences of living with PCa.” In lines 199-200 they write: “The aim of this study was to explore knowledge and beliefs of PCa among black South African survivors.” Both these statements contradict to previous claims that the goal of the study is to understand and explain screening behaviors.

Additional comments:

Line 105-106: “According to [24], people make meaning of their life experiences.” I would either expand this sentence and connect it better to their methodology and current research question or remove it. In its current form, it is a self-evident claim that does not contribute to the argument.

Lines 112-113: “The sampling choice was made because of its compatibility with hermeneutic phenomenological design [12].” More explanation is needed here to show how this sampling choice is compatible with phenomenological design.

Line 130: the word “transcribed” in this context seems to be incorrect. I assume the interviews were transcribed in the language they were conducted and then translated into English. It could also be helpful to add what the participants mother tongue was, especially considering that the authors later focus on linguistics differences and how they affect understanding of the disease, etc. Please provide more details here.

Line 133-134: “This method was chosen because of its compatibility with hermeneutic phenomenological inquiry.” This sentence needs to be removed or expanded to provide more details on how exactly this method is compatible with their own inquiry. Otherwise, this sentence does not add value to the manuscript. I suggest the authors avoid giving general description of a well-known methodology, but instead align this discussion with their own data and research questions.   

Line 158: “The participants (n=20; mean =76.2; SD =5.3) were 158 black men aged between 67 to 85 years” is reported previously in lines 155-156, no need to repeat it.

Lines 204-205 and Lines 207-208 are almost duplicate each other, please edit accordingly for clarity.

Lines 213-215: “Some of the participants in the study followed the traditional African belief system which holds that certain diseases can be transmitted through unforeseen supernatural forces (e.g., witchcraft, bad luck, wrath of angry ancestors, etc.)”  There is no mention about this argument in the results section. My suggestion is to add some examples to the results, if possible, as it sounds like an interesting and important theme in this context.

Lines 221-223: “According to [13], where no local translations or everyday equivalents exist, discussion of cancer (particularly prostate cancer) becomes difficult as people have no words of their own comfort to use.” My question here is about the word “cancer.” Does it mean that there is no word for “cancer” in the language these men speak? If not, what word did they use to talk about it? I think it is a very important to present more detail here. More details on language use will also be helpful to better understand the discussion in lines 224-233. It seems to be very interesting, and I also suggest focusing a bit less on the previous studies but bring in the new findings reported in this study and how they contribute to better understanding these men’s experiences.

Line 246: There was nothing in the results or discussion about these men believing or not believing that they are at risk for developing cancer. I suggest the author either add more details about this theme or rewrite the conclusion focusing on the findings they report in the manuscript.

Also, just a note: I would not refer to a small sample size as a limitation. It is a good sample size for a qualitative study, and its value is not determined by the sample size.

Minor style / grammar / spelling notes:

Line 41: “Studies” and line 55: “Screening” – should not be capitalized.

Line 73, 78: define LIMC or is it LMIC (line 37)?

Lines 89-90: “The cultural worldview of individuals is rooted in the values, beliefs and behaviours of their ethnic population” The structure of the sentence is not clear, I would suggest re-writing it: ““The cultural worldview of individuals is rooted in the values, beliefs, and behaviours of ethnic groups they belong to.”

Line 133: Define IPA? (it’s been defined in the abstract, but I would repeat it here to make it easier for the readers).

Line 236: “…cancer is an incurable (cancer fatalism.)” Remove “an”?

Line 250: “they belief” should be “they believe

Comments on the Quality of English Language

There are some minor issues, but easy to read and understand 

Author Response

Attached, receive the table of corrections.

Round 2

Reviewer 1 Report

Comments and Suggestions for Authors

My concerns have been addressed adequately.  Thanks!

Reviewer 2 Report

Comments and Suggestions for Authors

Thank you for the edits you've made to the manuscript.